# The social implications of participant choice on adherence to Isonaizid Preventive Therapy (IPT): A follow-up study to high completion rates in Eswatini

S. W. Grande[1,2]*, L. V. Adams[1], T. S. B. Maseko[3], E. A. Talbot[1], D. deGijsel[1], J. Mikal[2], Z. Z. Simelane[3], A. Achili[3], M. Mkhontfo[3], S. M. Haumba[3]

1 The Geisel School of Medicine at Dartmouth, Hanover, NH, United States of America, 2 University of Minnesota, School of Public Health, Minneapolis, MN, United States of America, 3 University Research Co. LLC (URC), Mbabane, Eswatini

* stuart.s.w.grande@dartmouth.edu

**Data Availability Statement:** All relevant data are within the paper and its Supporting Information files.

## Abstract

### Background

Eswatini (formerly Swaziland) has one of the highest rates of TB and HIV co-disease in the world. Despite national efforts to improve service delivery and prevent TB and HIV transmission, rates remain high. A recent prospective, observational study of integrated, patient-selected IPT delivery showed extraordinary improvements in IPT adherence, running counter to previous assumptions. This prompted the need to understand contextual and unseen study factors that contributed to high rates of adherence.

### Objective

To investigate high rates of IPT adherence rates among people living with HIV who participated in an observational study comparing modes of IPT delivery.

### Methods

Community-based participatory research guided the development of in-person administration of semi-structured questionnaires. Observational and field note data were analyzed. Qualitative data were analyzed using content analysis.

### Results

We interviewed 150 participants and analyzed responses from the 136 who remembered being given a choice of their IPT delivery method. Fifty-seven percent were female and the median age was 42. Nearly 67% of participants chose to receive facility-based IPT. High rates of self-reported IPT treatment adherence were linked to four key concepts: 1) adherence was positively impacted by community education; 2) disclosure of status served to empower participant completion; 3) mode of delivery perceptions positively impacted

**Funding:** Support for this work was provided by the TB CARE II project, which is funded by the United States Agency for International Development (USAID) under Cooperative Agreement Number AID-OAA-A-10-00021. The project's prime recipient is the University Research Co, LLC (URC), Chevy Chase, MD, USA, and the Geisel School of Medicine at the Dartmouth Section of Infectious Disease and International Health, Hanover, NH, USA, is a sub-recipient. The funding source for this work did not influence the study design, interpretation of data, writing of the manuscript, or the decision to submit the manuscript for publication. The funder provided support in the form of salaries for authors SWG, LVA, TSM, EAT, DG, ZZS, AA, MM, SMH, but did not have any additional role in the study design, data collection and analysis, decision to publish, or preparation of the manuscript. The specific roles of these authors are articulated in the methods sections and can be clarified here. SWG, LVA, EAT, and SMH contributed to the design of the project methods. SWG, LVA, EAT, DG, ZZS, AA contributed to data collection, writing, and analysis. All authors contributed to completion of final manuscript.

**Competing interests:** This work was undertaken within the resources available to the authors via their institutions. The authors declare no financial or other conflicts of interest. Support for this work was provided by the TB CARE II project, which is funded by the United States Agency for International Development (USAID) under Cooperative Agreement Number AID-OAA-A-10-00021. The project's prime recipient is the University Research Co (URC), Chevy Chase, MD, USA, and the Geisel School of Medicine at the Dartmouth Section of Infectious Disease and International Health, Hanover, NH, USA, is a sub-recipient. Any affiliation with home institutions does not alter our adherence to PLOS ONE policies on sharing data and materials. The authors declare no financial or other conflicts of interest.

adherence; and 4) choice of treatment delivery seen as helpful but not essential for treatment completion.

## Discussion

Achieving higher rates of IPT adherence in Eswatini and similar rural areas requires community-engaged education and outreach in coordination with care delivery systems.

## Background

Adult TB/HIV co-disease rates in Eswatini (formally Swaziland) are among some of the highest globally [1], and with rapidly evolving drug-resistant strains of TB comes an increased risk for people living with HIV (PLWH). While the country is moving quickly towards UNAIDS treatment targets of 90-90-90 (90% aware of status, 90% on HIV treatment, and 90% virally suppressed), there are concerns that immune compromised individuals remain at a much higher risk of acquiring tuberculosis. Impressive gains in the fight against HIV and TB demonstrate how local systems can coordinate to mitigate international epidemics; however, due to social and behavioral factors and speed of disease progression, screening rates for TB are much lower than necessary, limiting health systems' ability to reduce new TB infection [2; 3].

Isoniazid Preventive Therapy (IPT) is the recommended course of treatment to reduce the incidence of TB in PLWH. The challenge, according to the WHO, is ensuring continuation of care to reduce high rates of TB among PLWH who started ART [4]. Despite the adoption of a government program to provide IPT to tuberculosis child contacts and PLWH in 2011, by 2014 less than 10% of eligible individuals were receiving IPT [5]. Recent evidence points to the importance of self-selection of treatment delivery as a potential game-changing intervention to promote adherence of IPT [6], yet inadequate management of diagnosis and treatment are not having the desired success because of inadequate patient education, incomplete care coordination, and lack of healthcare communication [7; 8; 9].

Eswatini rates of IPT treatment completion are low; at 32% [5] they are even lower than the suboptimal rates of 40–50% reported in other settings [10]. Clinical definitions of adherence mean successfully taking medications >80% of the time, the right way at the right time. Challenges of adherence for IPT among PLWH reflect issues consistent with previous studies and include inadequate patient education and awareness of treatment. This has led researchers to suggest that many at high risk for TB are not aware of IPT as a viable treatment option [11]. Further, what these findings underscore are the persistent environmental challenges of acquiring medication and managing the demands of home, work, and health among PLWH and others needing preventive TB treatment.

While newer models of health communication have emerged in the last decade to promote adherence, very few studies have looked to modify treatment strategies with applying novel delivery strategies with TB, HIV, or IPT. In response, the authors conducted a larger observational study [6] to examine the impact of offering patients different modes of treatment delivery for receiving their IPT and its impact on adherence. Of three models of IPT delivery examined (community, facility, and peer-support), a majority selected a facility-based model and none selected the peer-support model. Although findings showed that self-selected mode of IPT delivery contributed to higher rates (94.8%) of adherence, there was little explanation for these observed rates [6]. As a result, the authors conducted a qualitative study to identify these unexplained factors.

## Methods

This study was approved by the Eswatini National Health Research Review Board, the University Research Co., LLC's Institutional Review Board (IRB) (USA), the IRB for Human Subject Research at Baylor College of Medicine and Affiliated Hospitals (USA), the IRB for Baylor College of Medicine Children's Foundation-Eswatini (Eswatini), and Dartmouth College's Committee for the Protection of Human Subjects (USA). Verbal consent was obtained from participants.

### Study team

The research team included a US-based medical sociologist, and three Eswatini researchers: two University Research Council (URC) research assistants, and a project manager, all of whom were supported by project principal investigators in the US and Eswatini. URC has been working on TB and HIV programs in Eswatini for decades. As such, the two research assistants and project manager were from the region and familiar with the TB and HIV care systems as well as local culture, contexts, and language. These team members were critical to the data collection process from both a socio-cultural perspective as well as from a researcher perspective. They were able to easily navigate local dialects, habits, and were especially considerate of clinical needs at each of the five locations.

### Study population (Selection and enrollment)

Stratified purposeful sampling was used to recruit 150 from the original 908 participants in our previous study. A sample matrix informed recruitment process based on five enrollment sites, mode of delivery, gender, and age. Eligibility for this study required participants to have fully adhered to a course of IPT delivered as part of our earlier study [6]. Three IPT delivery models included a community-based model, peer-support model, and a facility-based model. The community-model had clinic staff visit patients at home where they provided TB screening, adherence monitoring, and ART and INH refills. The facility-model asked patients to receive their screening, monitoring, and refills at a clinic. The peer-support model had patients join a peer-support group with a clinic staff member where screenings and monitoring were done and refills provided.

The local research team along with clinic managers across the five sites worked together to recruit participants. Initial contact was made by a clinical manager to identify and determine participant interest. In consultation with clinic managers, researchers contacted selected participants to coordinate a date and time for an interview. Once provisionally enrolled, members of the research team would meet with participants to obtain consent by describing the study or obtaining verbal assent from a caregiver or parent.

The site locations were chosen based on urban, rural, and peri-urban geography. Selection of the five study sites is previously described [6]. Two sites were rural, being over a 1.5 hour drive from the capital Mbabane. The other three locations were urban or peri-urban, and were located closer to a larger clinic/hospital. The research team who conducted the interviews were responsible for scheduling interviews and travelling by car to conduct the interview.

### Data collection

Data were collected between June and October of 2017. The research team designed an interview protocol to capture patient responses to a semi-structured questionnaire (S1 and S2 Files). Several topic domains were considered to help characterize the questionnaire, which were based on observations from the previous study and supported by professional

experiences of the team. These included participants' interaction with the healthcare system, perspectives receiving care while in the IPT study, perceptions of choice of IPT treatment, perceived challenges to and support for IPT use in Eswatini, and individual reactions to being in a research study.

Eswatini researchers conducted interviews in person with participants at each of the five healthcare facilities. The time and date of interview were scheduled to coincide with participants' usual HIV clinic appointment. All interviews were conducted in person, after obtaining written consent. In situations where a participant was unable or unwilling to meet in person, a phone survey was offered. For participants interviewed remotely, we obtained a waiver for written consent as permitted by the ethics review committee of record.

The formal process for conducting interviews was as follows: the team arrived at the clinic and met with clinic managers; the clinic manager introduced the interviewer to the patient; the researcher obtained written consent, and then conducted the interview in a private room or private outdoor space at the clinic. The interviewers usually conducted the interviews individually, where they would read the survey questions to the participants and take notes, adding comments to supplement qualitative questions. All the interviews were conducted in siSwati and in every case the semi-structured questionnaire was in both siSwati and English. The research team took notes in both English and siSwati, depending on the level of literacy of the participants and their preference for viewing and confirming written notes. Local team members would discuss and debrief survey findings each day they traveled to the sites to collect data. These reviews were summarized in bi-weekly updates shared with all research partners on a password protected file-sharing service.

Self-reported data from the questionnaire and additional open-ended questions were captured on paper for all participants. These data were then translated from siSwati to English in the database at the completion of every interview trip. At times when there was confusion around language or what was written on the survey, the team would meet and discuss until consensus was reached. Team members kept track of interviews, locations, and participants through a number classification system. This allowed international research partners to review data on a protected database with ease.

## Analysis

Interviewers transferred the survey data from paper files to a shared database on a password protected computer at URC office in Mbabane. Qualitative data were assessed via shared platform where the team coordinated a data reduction process based on an established multi-step thematic analysis described by Braun and Clarke [12]. Our approach was informed by a broader theoretical framework in that we utilized simultaneous data collection and analysis as well as a method of constant comparison [13]. As Braun and Clarke have suggested, the flexibility of thematic analysis permitted the parallel use of both inductive (data driven) and deductive (theory driven) approaches [12]. This flexibility permitted us to address certain concepts we knew *a priori* to be present as well as those we knew little about, the uncertainties around adherence behaviors.

The first phase of our analysis process was to discuss and review the data as it was being collected. As a team we then began to construct draft codes to describe what we were observing in the data. This process of identifying codes and revising codes occurred at multiple time points over the course of the study. Each time the team met in Eswatini, the data would be reviewed as a team, new codes would be introduced (where necessary) and old codes would be consolidated. The process of clarifying groups of codes and drafting themes would involve engaging each team member in discussions about survey responses, informal discussions with

participants, and cultural reflections by team members about their own experiences collecting data. At the end of each data collection trip the local researchers would meet to clarify their experiences and compare notes. Every three months, during the data collection process, the team would have video "check-in" sessions using a video chat software to speak with a more senior qualitative research member about progress and theme developments. As data were discussed, codes grouped together, and initial themes formed.

## Results

From the 150 selected, 14 participants were excluded as they did not recall being offered a choice of treatment. From a total of 136 participants sampled (Table 1), a representative group of 28 questionnaires was selected based on study sample matrix, as described below. These were assessed thematically over time as data were compiled using data reduction strategies of coding, theme generation, and memoing. The team's thematic analysis drawn from the 28 documents were then used to validate findings from the other 108 questionnaires. This form of validation was adopted as recording was unavailable due to resource limitations.

### Main findings

Factors that contributed to participant self-reported adherence to IPT could be characterized by four themes (Table 2): 1) Adherence to IPT was positively impacted by broader education that sensitized communities to value of IPT and risks of untreated TB; 2) Disclosure of health status and IPT enrollment acted to empower participant completion of IPT; 3) Perceptions regarding specific mode of delivery contributed positively to participant adherence; and 4) Choice of treatment delivery seen as helpful, though not essential for treatment completion. These four themes suggest that a broader means to achieving higher rates of self-reported adherence for IPT include education, supporting health status disclosure, finding solutions that fit people's lives, and giving people an opportunity to choose for themselves the delivery mode of their treatment.

### Self-reported adherence was positively influenced by indirect and direct forms of education

We define education as a process by which participants involved in the study supported a broader sensitization of communities about the value of IPT as well as the risks of untreated TB. Education includes being offered information and knowledge about IPT or TB and includes expressions of social support characterized as appraisal (useful for self-evaluation),

**Table 1. Participant demographics.**

| Variable | N (n = 136) | % |
|---|---|---|
| Sex | | |
| Female | 86 | 57.3 |
| Age distribution (years) | | |
| 0–17 | 22 | 14.7 |
| 18–35 | 45 | 30.0 |
| 36–49 | 51 | 34.0 |
| ≥50 | 32 | 21.3 |
| Final IPT delivery model | | |
| Community based | 50 | 33.3 |
| Facility Based | 100 | 66.7 |

**Table 2. Themes associated with improved adherence to IPT treatment.**

| Themes | Description |
|---|---|
| Self-reported adherence to IPT **by indirect and direct forms of education** | Education was applied broadly to include experiences that reflect the training of staff and clinicians in motivational interviewing, staff-supported peer support, and material for IPT delivery, which all led to greater information exchange with participants in study. |
| Disclosure of status and enrollment acted to empower participant completion of IPT | Disclosure in this case reflected stories of participants who shared their study experience, disease experience, and health status with friends, family, and employers. |
| Perceptions regarding specific mode of delivery contributed positively to participant adherence | Attitudes and beliefs varied across participants regarding mode of delivery: Facility-based, Community-based, and Peer-support model. Perceptions included ideas about quality of care, trust, and previous experience. |
| Choice of treatment delivery seen as helpful, though not essential for treatment completion | Being offered a choice for mode of care delivery was linked to having an option, being asked to choose, and autonomy. |

informational (advice), and instrumental (service or direct aid). A level of awareness reported by participants, seemed to act as a contributing factor to sensitizing patient communities to benefits of IPT for TB prevention. In other words, the way people communicated about their illness with family and friends seemed to indicate an uncommon awareness of symptoms, risks, and treatments.

Participants in the study reported that they learned informally from others involved in the previous study about the importance of IPT and the connection between HIV and TB. Another example of this knowledge transfer occurred when a woman shared the value of her learning about taking ART medications along with IPT and how her husband also supported her,

*"I would look at the time, when I take ARTs I would also take IPT and my husband would also remind me." (female, 36–49 years, facility-based model)*

The presence of community health workers enabled the informal exchange of health-related information by promoting positive health behaviors and strategies. Participants reported that this was useful for learning up-to-date details on how to care for themselves and others. In this broad way, education affiliated with ART medication was also linked to adherence, as demonstrated by another participant commenting here about what she learned from her HIV education,

*"I learned from the lesson we were given on ARTs, the importance of taking your treatment on time and completing your medication." (female, 18–36 years, facility-based model)*

How participants learned from site staff was reported as a key driver of IPT completion. Being alerted indirectly or directly to the risks of TB was recognized across interviews as being central to the success of IPT adherence. As TB and HIV are thought to have very similar symptoms, education at the clinics and in the communities was helpful for participants to differentiate between the two. Informal conversations with non-participants clarified for members of the research team that stigma associated with HIV symptoms are often conflated with symptoms of TB—loss of weight, skin coloring, coughing, and going to the clinic often for medications. Sensitizing community members, one family member at a time, appeared to be a consistent refrain. One participant made it clear to our team that educating friends was important.

"...[I told friends] I was raising awareness amongst them since I am a patient of TB so I wanted them to know the about the development in treatment." (male, 36–49 years, community-based model)

For other participants, educational messages described how they gained knowledge through informal exchanges with friends. Further, the way participants described seeking help for TB as a PLWH demonstrated they understood that IPT was beneficial for the individual as well as for family and friends. We observed this in one participant who made the critical point that having HIV made them more susceptible to acquiring TB.

"If positive, there is a higher risk of opportunistic infections including TB. So better prevent it.

Because if one is HIV positive, they are prone to TB." (female, 18–35 years, facility-based model)

## Disclosure of status and enrollment acted to empower participants to complete IPT

Personal relationships where participants felt comfortable enough to disclosing their status appeared central for treatment completion and IPT adherence. Participants reported disclosing through sharing their study experience, disease experience, and health status with their friends, family, and employer. Disclosure helped in the transmission of basic support, reminding about appointments and taking their medications regularly. When asked if they shared being in the study, the vast majority did, and chose to disclose in order to gain support to help them finish medications.

"I didn't want them to be surprised by the car coming home, because the IPT pills were brought home; and because I had already disclosed my [HIV] status." (female, 36–49 years, community-based model)

In these instances, we refer to personal relationships as any involving friends or family members, as well as neighbors and co-workers. We also observed a subtle difference in the disclosure of status between those who are seen as "inside" the household and those who are seen as "outside" the household. Here we have an example of someone who shared with their neighbors because they understood they needed more help, perhaps something they could not have achieved without disclosing.

"They were all aware of the poor conditions I lived under, so they helped me with food parcels so that I can take my pills daily."(female, 36–49 years, community-based model)

When asked about talking with others about being in the IPT study, participants appeared comfortable telling someone very close to them like a spouse or sibling. They could share the burden of remembering to take medication. In this way we observed better adherence in people who openly commented on the benefits and risks of disclosure, as those who disclosed were able to more successfully ask for help, share feelings of fear and sadness, and receive words of encouragement from those who matter most. One participant emphasized the importance of sharing with her sister.

"It very helpful since I was staying with my sister then, she would remind me encourage me to take my medication, even when I was no longer staying with her sometimes she would call

*and ask me if I am taking my medication accordingly and asked if I am eating the right diet."*
*(female, 18–35 years, facility-based model)*

While some participants commented that being honest with friends and family was important to reduce any potential surprises, the majority of participants who disclosed did so as a means of gaining support and sympathy. One participant's comment exemplifies the value of disclosing to close family members,

*"My husband and my mother, I wanted her to remind me since he is my treatment supporter even for my ART treatment." (female, 18–35 years, facility based-model)*

The act of disclosing status to employers appeared consistent with comments about disclosing to family. Sharing status with important others, including employers, strongly suggests that disclosure served to open opportunities that supported adherence and treatment completion.

## Perceptions regarding specific mode of delivery contributed positively to participant adherence

We define perceptions among participants as attitudes and beliefs regarding mode of delivery: facility-based, community-based, and peer-support model (described previously Adams et al, [6]). Perceptions included ideas about quality of care, trust, and previous experience. The facility-based model of care delivery was the default choice for a majority of participants, as many felt services in the community were substandard. One participant described their experience,

*"The RHMs are lazy, they are not active so any community based services are not good."*
*(female, facility-based model)*

We observed that perceptions of sub-standard community-based services often motivated people to seek care at facilities which were perceived as having higher quality care, while also offering anonymity and access to a doctor. Many participants spoke about the comfort they had with facilities, as they represented a more established and reliable treatment option. Participants have a sense that if they arrive sick and need help, they know they will be seen at a facility. Consequently, the facility model was seen has being more trustworthy than the community model,

*". . . but sometimes there are some stock- outs in some medication and not having adequate water supply will result in healthcare workers not working on that particular day which is inconvenience for patients." (female, community-based model)*

Participants expressed different preferences that influenced decision-making. This pointed to the need for determining ability to seek care at the facility, access to transportation, resources to pay for transportation, health / ability to travel long distances, and considerations of child care. As one participant suggested, the choice to receive care at a facility was made easier due to their access to resources,

*"Yes, I could do things based on my circumstances and financial abilities." (male, 36–49 years, facility-based model)*

While the facility-based model was preferred, in many circumstances, participants commented on the value of having the choice to receive treatment in the community as necessary and key to their treatment completion. One participant described the challenge of not having transportation access as being much less of a problem because there was a second option available to receive treatment.

> "*I thought it would be convenient because I don't have bus fare to the clinic.*" *(N/A gender, >50years, community-based model)*

Other participants also reflected on the value of convenience. In many circumstances, things emerge in day-to-day life that are unpredictable, and having a choice to get medications locally was viewed as a real benefit. One participant described switching from a facility model to a community model after hearing from a HCW that it would be easier for the person to receive their treatment in the community.

> "*HCW suggested that it was convenient for patient to switch to community due to disability.*" *(female, facility-based model)*

Unquestionably, participants were able to complete their treatment regimen based on having multiple options that fit with their lifestyle and the natural uncertainty of managing day-to-day living.

## Choice of treatment delivery seen as helpful but not essential for treatment completion

In the scope of patient experiences in Eswatini, models of care delivery reflect a more paternalistic approach to patient communication. Consequently, when asked about having a choice or having the opportunity to choose how and where they would receive their IPT protocol, many participants positively remarked that having a choice made a positive impression. A deeper look into these responses, it became clear than many participants had never been offered a choice of treatment prior to the IPT study. While a majority of participants chose to receive their treatment at regional facilities, this preference reflected a widely held sentiment that facilities are respected and provide high quality care. Further, facilities were seen as having fewer stigma-related challenges, while communities were higher risk for unintentional status disclosure.

> "*I was avoiding discrimination and criticism from my community as I know stigma is associated with the diseases.*" FBQ3 (Field note quote)

Participants who went to facilities also felt that receiving treatment at facilities there meant they would be able to get other care (e.g., health questions answered) at the same time while not having to go to multiple places.

> "*We do have health programs amongst people living with HIV, getting to facility is much easier and we are taught about condoms and other related health issues.*" GQ6 (Field note quote)

Beyond whether it was a community or facility, the choice of mode of treatment delivery was viewed by a majority of participants as an important feature of the study that helped them adhere to treatment. One participant made the point that choice helped overcome the burden

of stigma by empowering their decision to seek their medications at a location that was most appropriate.

> (Community) *"Yes, [I thought choice helped me] I picked them from clinic where I was comfortable at, and it became easy for me to do as per the nurse's advice without being intimidated by anything." (female, 18–35 years, community-based model)*

> (Facility) *"Yes. . . I would take them [IPT pills] without fear of stigma because I was comfortable with my choice." (male, unknown age, facility-based model)*

It became clear in the analysis process that participants expressed strong feelings about being able to make a choice for treatment delivery site (mode). For many, the basic idea of being offered a choice, and then being given the opportunity to choose for themselves, reflected a new kind of thinking. Drawn from a field note regarding the value of nurses and their connection to participants, there was a recognized sentiment among some participants that being offered a choice showed that the nurses cared. Consequently, many of the responses support the notion that choice led to feelings of empowerment and feeling connected with clinic staff.

## Discussion

A clinical study conducted by authors demonstrated notably high rates of IPT adherence. We concluded that more information was required to determine what factors promoted this high rate of adherence. This finding was of particular interest given recognized low rates of adherence in Eswatini compared to other sub-Saharan countries [14; 15]. One study suggested that discontinuation of IPT for PLWH was due in part to negative reactions to treatment [16] while others indicate nonadherence was due to misunderstanding risks and benefits [17]. Other potential causes for low rates of adherence have been attributed to patients not being fully informed about the pros and cons of treatment, and consequently patients not having enough information/confidence to complete their treatment given their reaction to medication [18].

Our study found four main themes that characterized improved treatment adherence (Fig 1. As shown in the consolidated figure, *treatment adherence* was positively impacted by community education, *disclosure* of status served to empower participant completion, *mode of delivery* perceptions positively impacted adherence, and *choice* of treatment delivery seen as helpful but not essential for treatment completion. Given the integrated nature of *choice* and *mode of delivery*, there is enough contextual overlap to compel parsimony in our model; yet, there is sufficient thematic difference to merit unique attribution of content. A consistent thread throughout was the perceived value of choice. While viewed as critical for many, and also not essential for treatment adherence, having a choice appeared to promote autonomy and empower participants to make necessary, preference-based decisions about where to receive their treatment. Our findings confirm that patients who understood the risks of treatment and were aware of the side-effects, were able to gain support needed to remain in the study and complete their treatment. Positive perceptions among communities about the risks and benefits of treatment to promote adherence has been shown to overcome stigma and appears consistent with our findings [19].

While the idea of patients being more likely to adhere to treatment if they understand the risks is not new, there is strong theoretical backing from the health education literature, particularly around health behavior theory, which shows that individuals tend to construct beliefs and motivations based on perceptions of severity of illness and susceptibility to acquiring illness [20]. Other health behavior theory, particularly the reasoned action approach, which focuses on how attitudes, social norms, and social pressures shape behavioral intentions, suggests that

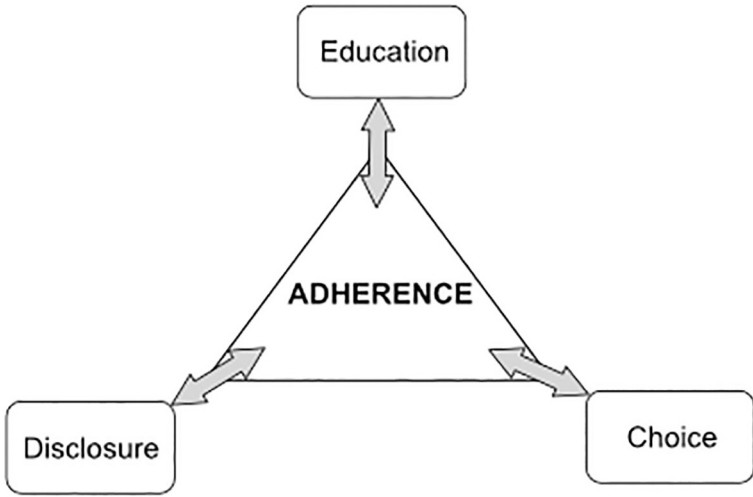

**Fig 1.**

adherence may be influenced by a confluence of attitudes and normative considerations [21]. In our findings, these behavioral determinants were also raised by participants as being indicative of barriers to adherence; however, the difference appeared to be participants who took a risk and disclosed their status were able to overcome many of these barriers to receive IPT.

Research on adherence describes how important sharing information with others can be for helping people feel connected and responsible. We found that disclosure of status served this purpose. While this is consistent with other research describing the value of family and social support factors [22], our findings identify the act of disclosure as a means to empower and activate social support systems. Those who were able to complete their treatment were supported by those closest to them, which include friends, family, and employers. This also recognizes the value of overcoming stigma in the community as a way to support better public health. Much has been written on the negative effects of nondisclosure or keeping status a secret [8], while other more recent findings state that a majority of patients that disclosed status to family or household did so in order to receive support/care [23]. Following the natural phenomenon of participants supporting others in these communities to share their status, there is evidence to suggest that people are more likely to take their medicines regularly, find ways to leave work and pick up medications, and receive emotional support when needed.

Our findings also illustrate the importance of education as a way to empower patients to be better prepared and activated to make decisions about their treatment. We found in our analysis that participants learned about the importance of IPT from multiple sources including clinic staff, who received specific training about the study, and from others who may have experienced TB or other forms of preventative treatment for TB. We know that information can be valuable when shared through the right channels, and our findings confirm that in all settings, especially when dissemination of information can be a challenge, providing adequate upfront education to clinic staff and study participants served to "sensitize" others in the community to the benefits of IPT.

## Limitations

While the findings from this work are informative and novel to the context of TB care and prevention, based on some methodological limitations our conclusions should be read with

caution. As our findings report qualitative results of open-ended responses, we recognize that some comments were biased towards those who were more confident in formal conversations and still others who were more confident having informal conversations with research assistants. In other scenarios, participants responded to surveys that were in siSwati and back translated into English. While these translations were checked by members of the research team and native siSwati and English speakers, it is possible that some participants may have misunderstood the intention of some concepts like the difference between community health services and health workers or other concepts like access to care and availability of services. Interview data were captured by open-ended survey response, and not recorded, which may have limited verbatim responses; however, recognizing these limitations, research assistants checked understanding with participants periodically to confirm or clarify these concepts. With any qualitative data there is a limit to generalization and interpretation. Some consideration of selection bias is needed as our study captured responses of those who completed treatment, purposive sampling to identify 150 participants, and finally those who were interviewed at the facility. These decisions reflect an intentional design to capture programmatic feasibility and impact among patient-selected IPT delivery model, which did not favor all patients. These findings reflect very specific responses to care received in Eswatini for TB and HIV and the delivery and receipt of IPT.

## Future considerations

Recommendations for improving IPT delivery include more focused and scaled community education sessions. At the conclusion of this project, our team designed an IPT toolkit that has been disseminated widely at international TB meetings and is freely available via the TB CARE II website (https://tbcare2.org/wp-content/uploads/2019/01/IPT-Delivery-Toolkit-web.pdf). The research process, working in partnership with a local NGO research team, building clinical communication strategies, offering choice of mode of treatment, and keeping in close contact with partnering clinics were essential aspects to this work. Our findings, while very specific to Eswatini, reflect basic strategies of work that can be replicated elsewhere when tailored appropriately to country context and culture.

## Conclusion

Our qualitative findings identified key features for improving rates of adherence of IPT in Eswatini, which include providing clear and authentic education of IPT services and benefits, supporting participants to disclose status, and offering patient choice for mode of IPT delivery. Based on these findings, recommendations for improved services for managing and preventing TB among PLWH are strongly linked to providing autonomy and empowering individuals to make treatment decisions that match their expectations and lifestyles.

## Supporting information

**S1 File. Questionnaire (English).**
(DOCX)

**S2 File. Questionnaire (Siswati).**
(DOCX)

## Author Contributions

**Conceptualization:** S. W. Grande, L. V. Adams, E. A. Talbot, S. M. Haumba.

**Data curation:** S. W. Grande, L. V. Adams, T. S. B. Maseko, E. A. Talbot, D. deGijsel, Z. Z. Simelane, A. Achili, M. Mkhontfo.

**Formal analysis:** S. W. Grande, L. V. Adams, T. S. B. Maseko, E. A. Talbot, D. deGijsel, Z. Z. Simelane, A. Achili.

**Funding acquisition:** L. V. Adams, E. A. Talbot, S. M. Haumba.

**Investigation:** S. W. Grande, L. V. Adams, E. A. Talbot, Z. Z. Simelane.

**Methodology:** S. W. Grande, L. V. Adams, T. S. B. Maseko, E. A. Talbot, D. deGijsel, J. Mikal, Z. Z. Simelane, A. Achili, M. Mkhontfo, S. M. Haumba.

**Project administration:** S. W. Grande, L. V. Adams, E. A. Talbot, M. Mkhontfo.

**Resources:** L. V. Adams, M. Mkhontfo, S. M. Haumba.

**Software:** T. S. B. Maseko.

**Supervision:** S. W. Grande, L. V. Adams, E. A. Talbot, S. M. Haumba.

**Validation:** L. V. Adams, T. S. B. Maseko, E. A. Talbot, D. deGijsel, J. Mikal, Z. Z. Simelane, A. Achili.

**Writing – original draft:** S. W. Grande, L. V. Adams.

**Writing – review & editing:** S. W. Grande, L. V. Adams, T. S. B. Maseko, E. A. Talbot, D. deGijsel, J. Mikal, Z. Z. Simelane, A. Achili, M. Mkhontfo, S. M. Haumba.

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
