## [Decision Letter · Decision Letter 0]

17 Feb 2020

PONE-D-20-02451

The social implications of participant choice on adherence to Isonaizid Preventive Therapy (IPT): A follow-up study to high completion rates in Eswatini

PLOS ONE

Dear Grande,

Thank you for submitting your manuscript to PLOS ONE. After careful consideration, we feel that it has merit but does not fully meet PLOS ONE’s publication criteria as it currently stands. Therefore, we invite you to submit a revised version of the manuscript that addresses all the points raised during the review process. Additionally, please note that while PLOS ONE considers qualitative and mixed-methods studies for publication, authors should use the COREQ checklist, or other relevant checklists listed by the Equator Network, such as the SRQR, to ensure complete reporting (http://journals.plos.org/plosone/s/submission-guidelines#loc-qualitative-research). In general, we would expect qualitative studies to include the following: 1) defined objectives or research questions; 2) description of the sampling strategy, including rationale for the recruitment method, participant inclusion/exclusion criteria and the number of participants recruited; 3) detailed reporting of the data collection procedures; 4) data analysis procedures described in sufficient detail to enable replication; 5) a discussion of potential sources of bias; and 6) a discussion of limitations.

We would appreciate receiving your revised manuscript by Apr 02 2020 11:59PM. To enhance the reproducibility of your results, we recommend that if applicable you deposit your laboratory protocols in protocols.io, where a protocol can be assigned its own identifier (DOI) such that it can be cited independently in the future. For instructions see: http://journals.plos.org/plosone/s/submission-guidelines#loc-laboratory-protocols

We look forward to receiving your revised manuscript.

Kind regards,

Katalin Andrea Wilkinson, PhD

Academic Editor

PLOS ONE

Journal Requirements:

2. Please provide additional details regarding participant consent. In the ethics statement in the Methods and online submission information, please ensure that you have specified whether consent was suitably informed. If your study included minors under age 18, state whether you obtained consent from parents or guardians.

3. Please include in your Methods section the date ranges over which you recruited participants to this study.

4. Please include a copy of the questionnaire used in the study in Siswati (if available), as Supporting Information.

We note that one or more of the authors are employed by a commercial company: University Research Co LLC 

6. We note you have included a table to which you do not refer in the text of your manuscript. Please ensure that you refer to Table 1 and 2 in your text; if accepted, production will need this reference to link the reader to the Table.

Reviewers' comments:

Reviewer's Responses to Questions

**Comments to the Author**

1. Is the manuscript technically sound, and do the data support the conclusions?

Reviewer #1: Partly

Reviewer #2: Partly

2. Has the statistical analysis been performed appropriately and rigorously? 

Reviewer #1: No

Reviewer #2: Yes

3. Have the authors made all data underlying the findings in their manuscript fully available?

Reviewer #1: Yes

Reviewer #2: Yes

4. Is the manuscript presented in an intelligible fashion and written in standard English?

Reviewer #1: Yes

Reviewer #2: Yes

5. Review Comments to the Author

Reviewer #1: This is an interesting and important study done looking at factors associated with positive adherence to IPT in ESwatini. The authors should be commended for using a qualitative methods approach to better understand this unexpected finding. There are, however, several concerns I have about the paper which I describe in more detail below:

1) The authors need to provide information about the different models of care offered in the study. While they reference the original Adams paper from the 2017 IJTLD in some places, this is not sufficient, and readers looking at this paper need to know what was offered within each model.

2) The introduction to this paper is too long and it needs to be condensed into 3-4 paragraphs that more concisely lay out the issues: Eswatini suffers from a high burden of both TB and HIV; IPT has been shown to save lives, but adherence rate are usually poor; in a larger study done by the group offering different models of IPT, high rates of adherence were seen; the authors decided to do a qualitative study to find out why.

3) The authors should describe in more detail the primary study that was done (just 2-3 sentences perhaps to give the readers more context).

4) In the methods section, the authors note that “The site locations were chosen based on proximity to clinics and hospitals”. This is understandable, but it may have biased the research significantly, as patients who had further to travel to the clinics might have preferred a different model of care delivery. This needs to be mentioned in the limitations section.

5) In the methods section, the authors note that they selected participants only among people who self-reported that they completed treatment. This is a nice example of focusing on a “positive” outcome. The authors also mention the potential for bias this sample has in their discussion section. However, they might want to spend more time discussing this. In qualitative research, one aim is to describe a range of experiences to fully describe an observed phenomenon and the exclusion of people who did not complete therapy limits this.

6) Under the data collection section, the authors report that the researchers “conducted interviews and completed surveys”. In looking through the interview guide, I see that there were some questions in which several options were offered and some that were more open-ended. However, from the guide it also appears that even when specific answer options were offered, there was also an opportunity for people to provide more details. Therefore, I would just refer to these as “conductive interviews.”

7) Is there a reason the interviews were not recorded and transcribed? This is generally the accepted methodology for qualitative research of this kind. The authors should specify why they did not do this and also the limitations of not recording the interviews.

8) It would also be helpful if the authors could describe in more detail the analysis method that was used. They note that they used a “thematic analysis”, and the process they describe seems to have both inductive elements (the open-ended questions) and deductive elements (the questions with specific options for answers that were based on previous research). They also should describe the theoretic framework more (i.e. they seemed to used grounded theory).

9) The authors need to note throughout the paper that they are describing self-reported adherence.

10) In the results section, sometimes the direct quotes from patients are presented in italics and set off from the text but in others, the quotes are just presented in the text. The authors should be consistent in how they are presenting this rich data.

11) The 4 themes presented are quite broad, and I am wondering if the analysis included identifying any subthemes? With some of the topics (i.e. education), sub-themes seem to be emerging in the different paragraphs. The authors should try to more formally assess sub-themes in all of their data as it would help make sense of some of the very broad findings. As an example, under education, there was formal “education” received in structured sessions with doctors/nurses/researchers. But then the authors also describe more “informal” education that happened during talks with neighbors (i.e. a participant reported being worried because his neighbor had TB and this prompted the participant to want to take IPT). The same is true in the section on perceptions regarding mode of delivery, where quality of care is explored along with waiting times, resources to get to clinics, etc.

12) In the section on disclosing status and enrollment in the IPT study, the authors report multiple positive experiences. This is interesting and may be due to the fact that most people in the study completed therapy. However, given the stigma often associated with disclosing TB status, this is somewhat surprising and merits further exploration. Perhaps people were more likely to disclose since they did not actually have TB (and thus did not fear transmission to others)? This topic should be discussed in more detail in the discussion section.

13) The four domains identified are interesting and important, but they also seem to have possible interactions with one another. For example, a experience with education from a clinician/nurse/study personnel may have given the patient more trust in the facility and thus they may have had more trust in the facility. While it is important to describe the 4 themes, the data analysis should also look at how these themes may have interacted with one another. Such an analysis is missing from this paper. A figure documenting the different themes, sub-themes, interactions between them, and their overall impact on IPT completion would greatly add to the value of this work.

14) Finally, in the conclusion, the authors note “alarming” rates of treatment adherence. I think the rates are surprisingly high, but the term alarming usually has a worrisome connotation to it, and I would recommend using a different term.

Reviewer #2: Thank you for the opportunity to review this interesting paper which explores factors that can contribute to high levels of adherence to IPT in Eswatini. Considering the persisting importance of TB in Eswatini and Sub-Saharan Africa, the paper investigates a relevant public health topic and is consequently of general interest. The paper clearly situates the problem and research question, That said, the manuscript has several shortcomings.

1) The manuscript is very long and can be substantially shortened.

2) While this is a qualitative study, the findings contain long descriptions of the context and analyses of patients' experiences and perceptions. However, very few quotes are used to support these analyses for the 4 main themes. In addition, few of the quotes that are used support the claims that are made. In light of this, I recommend to shorten the analyses in the findings, distill the absolute essence for each theme and support it with patients' words.

3) For example, describing the waiting areas and waiting times distracts from the actual finding i.e. care at facilities is perceived as high quality and reliable.

4) Theme 3 describes perceptions regarding modes of delivery. These perceptions are well described but can be shortened. In addition, the observation that these perceptions contribute positively to IPT adherence is not sufficiently substantiated. I would recommend using more direct quotes to show the link.

5) Theme 1 describes different forms of education. Similar to the previous comment, this section can be shortened and needs more direct quotes to show the link between education and adherence.

6) I would recommend to simplify the language used, especially in the findings and discussion sections. As mentioned earlier, the topic and findings of this manuscript have an acute relevance for public health programming and as such could benefit from shorter, more succint and easier to read language to ensure it is accesible to a wide audience of healthcare workers, local goverment, programme implementers, policy writers, and researchers.

6. PLOS authors have the option to publish the peer review history of their article (what does this mean?). If published, this will include your full peer review and any attached files.

Reviewer #1: No

Reviewer #2: No

---

## [Author Response · Author response to Decision Letter 0]

1 Apr 2020

Response to Reviewers

Dear Editors and reviewers,

We would like to thank the reviewers for their thoughtful comments and helpful suggestions. We have worked hard to acknowledge each of their comments and think the contributions have improved the overall quality of the manuscript. We have shifted focus, based on reviewer comments, to the nature of our findings and how young people who had difficulty describing their symptoms and state of well-being were able to do so more effectively once they began using Genia. Further edits will show that we have recognized the limitations of this study and have addressed each of the reviewers’ comments as parsimoniously as possible.

Reviewer #1:This is an interesting and important study done looking at factors associated with positive adherence to IPT in eSwatini. The authors should be commended for using a qualitative methods approach to better understand this unexpected finding. There are, however, several concerns I have about the paper which I describe in more detail below:

Response: We appreciate the reviewers recognition of the work conducted and their identification of qualitative inquiry as a legitimate and rigorous method. We hope we’ve addressed their concerns sufficiently. 

Comment 1) The authors need to provide information about the different models of care offered in the study. While they reference the original Adams paper from the 2017 IJTLD in some places, this is not sufficient, and readers looking at this paper need to know what was offered within each model.

Response 1) This is an important addition and we have included the following language:

Three IPT delivery models included a community-based model, peer-support model, and a facility-based model. The community-model had clinic staff visit patients at home where they provided TB screening, adherence monitoring, and ART and INH refills. The facility-model asked patients to receive their screening, monitoring, and refills at a clinic. The peer-support model had patients join a peer-support group with a clinic staff member where screenings and monitoring were done and refills provided. 

Comment 2) The introduction to this paper is too long and it needs to be condensed into 3-4 paragraphs that more concisely lay out the issues: Eswatini suffers from a high burden of both TB and HIV; IPT has been shown to save lives, but adherence rate are usually poor; in a larger study done by the group offering different models of IPT, high rates of adherence were seen; the authors decided to do a qualitative study to find out why.

Response 2) We appreciate the suggestion and agree the introduction is too long. We have cut much of the language to enhance clarity. We have revised the entire introduction in response.

Comment 3) The authors should describe in more detail the primary study that was done (just 2-3 sentences perhaps to give the readers more context).

Response 3) We have added language to explain the initial study and give an explanation for the current study under review. Here is the added text:

In response, the authors conducted a larger observational study (Adams et al. 2017) to examine the impact of offering patients different modes of treatment delivery for receiving their IPT and its impact on adherence. Of three models of IPT delivery examined (community, facility, and peer-support), a majority selected a facility-based model and none selected the peer-support model. Although findings showed that self-selected mode of IPT delivery contributed to higher rates (94.8%) of adherence, there was little explanation for these observed rates (Adams et al. 2017).

Comment 4) In the methods section, the authors note that “The site locations were chosen based on proximity to clinics and hospitals”. This is understandable, but it may have biased the research significantly, as patients who had further to travel to the clinics might have preferred a different model of care delivery. This needs to be mentioned in the limitations section.

Response 4) We thank the reviewer for drawing our attention to this detail. We have revised the language to match our earlier protocol and include it below. We believe it more accurately reflects how the sites were chosen and why. As a result, we think the modified language does not require additional clarification in the limitations, yet, we due include language on selection bias in the limitations section.

The site locations were chosen based on urban, rural, and peri-urban geography

Comment 5) In the methods section, the authors note that they selected participants only among people who self-reported that they completed treatment. This is a nice example of focusing on a “positive” outcome. The authors also mention the potential for bias this sample has in their discussion section. However, they might want to spend more time discussing this. In qualitative research, one aim is to describe a range of experiences to fully describe an observed phenomenon and the exclusion of people who did not complete therapy limits this.

Response 5) We thank the reviewer for this insight, and agree that qualitative inquiry extends to deeper understanding of all cases and experiences. Noting this is a limitation, we simply do not have reports of those who did not complete treatment. As we note in the limitations section, this was due to our research question to understand high rates of adherence rather than experiences of participants. 

Comment 6) Under the data collection section, the authors report that the researchers “conducted interviews and completed surveys”. In looking through the interview guide, I see that there were some questions in which several options were offered and some that were more open-ended. However, from the guide it also appears that even when specific answer options were offered, there was also an opportunity for people to provide more details. Therefore, I would just refer to these as “conductive interviews.”

Response 6) We very much appreciate this comment and suggestion. We agree that the interview guide was used to support a discussion, a give and take, rather than an explicit call and response as might be likely with a survey. We have made the revision in the manuscript.

Comment 7) Is there a reason the interviews were not recorded and transcribed? This is generally the accepted methodology for qualitative research of this kind. The authors should specify why they did not do this and also the limitations of not recording the interviews.

Response 7) We thank the reviewer for their comment and agree that recordings are an accepted way to collect participants’ responses carefully and accurately. We had intended to have our researchers used recording devices but due to resource-related issues our team had to rely on written responses, note taking, and memo writing. Given the time researchers spent with participants and checked understanding, we were confident that data we collected accurately reflects participant knowledge, attitudes, and beliefs. Also, as we tried to state in our results, thematic analysis was validated using 28 questionnaires to support the findings from data collected from the 136 participants sampled. We acknowledge this limitation and have addressed this in the discussion.

Interview data were captured by open-ended survey response, and not recorded, which may have limited verbatim responses; however, recognizing these limitations, research assistants checked understanding with participants periodically to confirm or clarify these concepts. 

Comment 8) It would also be helpful if the authors could describe in more detail the analysis method that was used. They note that they used a “thematic analysis”, and the process they describe seems to have both inductive elements (the open-ended questions) and deductive elements (the questions with specific options for answers that were based on previous research). They also should describe the theoretic framework more (i.e. they seemed to used grounded theory).

Response 8) We thank the reviewer for this careful reading and thoughtful comment. To address this point we have added the following to the analysis section.

Our approach was informed by a broader theoretical framework in that we utilized simultaneous data collection and analysis as well as a method of constant comparison (Charmaz 2006). As Braun and Clarke have suggested, the flexibility of thematic analysis permitted the parallel use of both inductive (data driven) and deductive (theory driven) approaches (Braun and Clarke 2006). This flexibility permitted us to address certain concepts we knew a priori to be present as well as those we knew little about, the uncertainties around adherence behaviors.

Comment 9) The authors need to note throughout the paper that they are describing self-reported adherence.

Response 9) We have added the phrase “self-reported” periodically throughout the manuscript to attend to natural flow of writing. We also added self-reported to one of main themes to reflect this as a larger concept. 

Comment 10) In the results section, sometimes the direct quotes from patients are presented in italics and set off from the text but in others, the quotes are just presented in the text. The authors should be consistent in how they are presenting this rich data.

Response 10) We appreciate this comment and want to be as consistent as possible. We also recognize that some quotes were multiple lines and others were not, which may depend on the journal’s preference. 

Comment 11) The 4 themes presented are quite broad, and I am wondering if the analysis included identifying any subthemes? With some of the topics (i.e. education), sub-themes seem to be emerging in the different paragraphs. The authors should try to more formally assess sub-themes in all of their data as it would help make sense of some of the very broad findings. As an example, under education, there was formal “education” received in structured sessions with doctors/nurses/researchers. But then the authors also describe more “informal” education that happened during talks with neighbors (i.e. a participant reported being worried because his neighbor had TB and this prompted the participant to want to take IPT). The same is true in the section on perceptions regarding mode of delivery, where quality of care is explored along with waiting times, resources to get to clinics, etc.

Response 11) We thank the reviewer for taking the time to review the data with such detail and care. The inclusion of sub-themes was not part of the initial or follow-up data review process. As our team met multiple times over the course of several months and via web-chat, there were many decisions along the way to consolidate findings to focus on the main themes/factors contributing to ‘self-reported’ adherence. Sub-themes were viewed as taking away from the the broader impact of education and mode of delivery. In fact, as a research team we attempted at several points to break up the “broad theme” of education, but found it wasn’t easily differentiated into buckets, as individuals shared overlapping experiences and sentiments, suggesting there was a broader or more general form of education occuring, one based on lived experience.

Comment 12) In the section on disclosing status and enrollment in the IPT study, the authors report multiple positive experiences. This is interesting and may be due to the fact that most people in the study completed therapy. However, given the stigma often associated with disclosing TB status, this is somewhat surprising and merits further exploration. Perhaps people were more likely to disclose since they did not actually have TB (and thus did not fear transmission to others)? This topic should be discussed in more detail in the discussion section.

Response 12) We agree that the nature of this finding merits further investigation, and as part of our analysis we sought deeper understanding through the data. However, it came up that based on a potential sampling bias (pointed out by the reviewer) that this finding may be skewed, which we identified in the discussion. Also, while we don’t think we can speculate on potential fears of participants, our observations suggest that disclosure did occur and it appeared motivated by a desire for support, which was a finding supported by other recent research. 

Much has been written on the negative effects of nondisclosure or keeping status a secret (Gebremariam et al. 2010), while other more recent findings state that a majority of patients that disclosed status to family or household did so in order to receive support/care (Nyangoma et al. 2020). 

Comment 13) The four domains identified are interesting and important, but they also seem to have possible interactions with one another. For example, an experience with education from a clinician/nurse/study personnel may have given the patient more trust in the facility and thus they may have had more trust in the facility. While it is important to describe the 4 themes, the data analysis should also look at how these themes may have interacted with one another. Such an analysis is missing from this paper. A figure documenting the different themes, sub-themes, interactions between them, and their overall impact on IPT completion would greatly add to the value of this work.

Response 13) We very much appreciate the thoughts shared by the reviewer. We respectfully have to defend our substantive choice to keep the four main themes. These reflect hours and weeks of interdisciplinary discussion and debate across continents and cultures. While there is always room to recharacterize some of the relationships between themes and their contexts, my colleagues and I stand by these data as reflective of our participants and their lived experiences. As such we feel compelled to address some of the reviewers suggestions as they are useful and will improve the manuscript. 

We’ve added the following to the discussion:

Given the integrated nature of choice and mode of delivery, there is enough contextual overlap to compel parsimony in our model; yet, there is sufficient thematic difference to merit unique attribution of content.

We have also added the following figure to the manuscript to depict what we believe to be the interaction of each of the main themes, recognizing that choice and mode of treatment have some overlap.

Comment 14) Finally, in the conclusion, the authors note “alarming” rates of treatment adherence. I think the rates are surprisingly high, but the term alarming usually has a worrisome connotation to it, and I would recommend using a different term.

Response 14) We have changed to “notably”

Reviewer #2: Thank you for the opportunity to review this interesting paper which explores factors that can contribute to high levels of adherence to IPT in Eswatini. Considering the persisting importance of TB in Eswatini and Sub-Saharan Africa, the paper investigates a relevant public health topic and is consequently of general interest. The paper clearly situates the problem and research question, That said, the manuscript has several shortcomings.

Response) We appreciate the reviewers time and thoughts on the manuscript. We hope we have addressed the reviewers concerns in detail below.

Comment 1) The manuscript is very long and can be substantially shortened.

Response 1) We appreciate the reviewer’s comment and have reduced the introduction and descriptions in the manuscript substantially.

Comment 2) While this is a qualitative study, the findings contain long descriptions of the context and analyses of patients' experiences and perceptions. However, very few quotes are used to support these analyses for the 4 main themes. In addition, few of the quotes that are used support the claims that are made. In light of this, I recommend to shorten the analyses in the findings, distill the absolute essence for each theme and support it with patients' words.

Response 2) We thank the reviewers comments and have worked to refine the language in the results. We have also added a parsimonious figure to help mitigate potential areas of confusion. 

Comment 3) For example, describing the waiting areas and waiting times distracts from the actual finding i.e. care at facilities is perceived as high quality and reliable.

Response 3) We appreciate the reviewers comments and have made significant edits to the results sections, reducing narrative, and focusing on the key aspects of the findings. 

Comment 4) Theme 3 describes perceptions regarding modes of delivery. These perceptions are well described but can be shortened. In addition, the observation that these perceptions contribute positively to IPT adherence is not sufficiently substantiated. I would recommend using more direct quotes to show the link.

Response 4) We thank the reviewer for drawing attention to this gap. We believe that the extra language took away from core theme and have substantially cut sections from the theme narrative.

Comment 5) Theme 1 describes different forms of education. Similar to the previous comment, this section can be shortened and needs more direct quotes to show the link between education and adherence.

Response 5) We thank the reviewer for pointing to this gap in the manuscript. As we commented to the first reviewer, the themes generated reflect months of careful consideration and communication between researchers and community members. As a result we feel strongly that education functions as an umbrella theme to much of the work that contributed to participant adherence. We detail those contributions in the results, and do not provide sub-themes as there is meaningful overlap and nuance that needed to be stated. At the same time, we have produced a figure to support the four main themes and believe this figure provides a parsimonious. We hope this mitigates the reviewers worries about communicating key findings and meaning. 

Comment 6) I would recommend to simplify the language used, especially in the findings and discussion sections. As mentioned earlier, the topic and findings of this manuscript have an acute relevance for public health programming and as such could benefit from shorter, more succinct and easier to read language to ensure it is accessible to a wide audience of healthcare workers, local government, programme implementers, policy writers, and researchers.

Response 6) We very much appreciate the reviewers comments and have made substantial cuts to the narrative to improve the manuscript. Given the rich detail provided in the results, we were only able to cut in specific areas, but with a more streamlined introduction and new figure, we think we have sufficiently addressed the reviewers important concerns.

---

## [Editor Report · Decision Letter 1]

23 Apr 2020

The social implications of participant choice on adherence to Isonaizid Preventive Therapy (IPT): A follow-up study to high completion rates in Eswatini

PONE-D-20-02451R1

Dear Dr. Grande,

We are pleased to inform you that your manuscript has been judged scientifically suitable for publication and will be formally accepted for publication once it complies with all outstanding technical requirements.

With kind regards,

Katalin Andrea Wilkinson, PhD

Academic Editor

PLOS ONE
---

## [Editor Report · Acceptance letter]

8 May 2020

PONE-D-20-02451R1 

The social implications of participant choice on adherence to Isonaizid Preventive Therapy (IPT): A follow-up study to high completion rates in Eswatini 

Dear Dr. Grande:

I am pleased to inform you that your manuscript has been deemed suitable for publication in PLOS ONE. Congratulations! Your manuscript is now with our production department. 

With kind regards,

on behalf of

Associate Professor Katalin Andrea Wilkinson 

Academic Editor

PLOS ONE